# The Two Prevalent Genotypes of an Emerging Infectious Disease, *Deformed Wing Virus*, Cause Equally Low Pupal Mortality and Equally High Wing Deformities in Host Honey Bees

**DOI:** 10.3390/v11020114

**Published:** 2019-01-29

**Authors:** Anja Tehel, Quynh Vu, Diane Bigot, Andreas Gogol-Döring, Peter Koch, Christina Jenkins, Vincent Doublet, Panagiotis Theodorou, Robert Paxton

**Affiliations:** 1General Zoology, Institute for Biology, Martin Luther University Halle-Wittenberg, Hoher Weg 8, 06120 Halle (Saale), Germany; bigot.diane@gmail.com (D.B.); cej.jenkins@gmail.com (C.J.); vincent.bs.doublet@gmail.com (V.D.); panatheod@gmail.com (P.T.); 2Cuu Long Delta Rice Research Institute, Can Tho City 94709, Vietnam; vquynh@gmail.com; 3German Centre for Integrative Biodiversity Research (iDiv) Halle-Jena-Leipzig, Deutscher Platz 5e, 04103 Leipzig, Germany; andreas.gogol-doering@mni.thm.de; 4Technische Hochschule Mittelhessen, University of Applied Sciences, Wiesenstrasse 14, 35390 Gießen, Germany; Koch.Peter@gmx.de; 5Institute of Evolutionary Biology, University of Edinburgh, Charlotte Auerbach Road, Edinburgh EH9 3FL, UK

**Keywords:** pathology, virulence, positive single-strand RNA virus, DWV, genotype A, genotype B, *Apis rhadbovirus-1*, *Apis mellifera*

## Abstract

*Deformed wing virus* (DWV) is an emerging infectious disease of the honey bee (*Apis mellifera*) that is considered a major cause of elevated losses of honey bee colonies. DWV comprises two widespread genotypes: the originally described genotype A, and genotype B. In adult honey bees, DWV-B has been shown to be more virulent than DWV-A. However, their comparative effects on earlier host developmental stages are unknown. Here, we experimentally inoculated honey bee pupae and tested for the relative impact of DWV-A *versus* DWV-B on mortality and wing deformities in eclosing adults. DWV-A and DWV-B caused similar, and only slightly elevated, pupal mortality (mean 18% greater mortality than control). Both genotypes caused similarly high wing deformities in eclosing adults (mean 60% greater wing deformities than control). Viral titer was high in all of the experimentally inoculated eclosing adults, and was independent of wing deformities, suggesting that the phenotype ‘deformed wings’ is not directly related to viral titer or viral genotype. These viral traits favor the emergence of both genotypes of DWV by not limiting the reproduction of its vector, the ectoparasitic *Varroa destructor* mite, in infected pupae, and thereby facilitating the spread of DWV in honey bees infested by the mite.

## 1. Introduction

Honey bees (*Apis mellifera*) are globally the most important commercial pollinator of crops [1], as well as playing an important role in the pollination of wild plants [2]. Although the world stock of colonies has grown over the past half century in response to increasing pollinator-dependent agricultural production [3,4], honey bees in the temperate Northern Hemisphere have suffered increasing colony mortality in the past two decades [5,6,7]. Multiple explanations for elevated colony losses have been proposed, chiefly comprising: diverse forms of habitat degradation, parasites, pesticides, and interactions among all three factors [4,8,9]. Surveys of pests and pathogens of honey bees in the temperate Northern Hemisphere have highlighted the exotic and invasive ectoparasitic mite, *Varroa destructor*, which is a native ectoparasite of the S. and E. Asiatic honey bee *Apis cerana*, and the emerging infectious disease agent, *Deformed wing virus* (DWV), together being a major cause of elevated colony loss [10,11,12,13,14,15,16,17,18].

The invasion of honey bee populations by *V. destructor* leads to a greatly increased prevalence (proportion of infected individuals or colonies) and intensity of infection (viral titer per individual) of DWV [19,20]. DWV is a positive single-stranded RNA virus of the family Iflaviridae (order Picornaviruses) that has long been associated with *A. mellifera* [21]. Although transmission routes of DWV among honey bees are diverse [22,23,24,25,26] (reviewed in [27]), *V. destructor* is a highly efficient vector of DWV when it parasitizes honey bee pupal stages [28,29,30,31,32], leading to its emergence in the presence of *V. destructor*. Although DWV may also replicate within *V. destructor* [24,30,31,33] (summary in [34]), enhancing its transmission, other evidence argues against the within-mite replication of DWV [32,35]. Regardless of whether *V. destructor* acts as a mere vector of DWV or whether the virus also replicates in the vector, the direct effects of DWV *per se* on honey bees have been difficult to separate from those of feeding by *V. destructor* on host pupae during metamorphosis, but include deformed wings, shortened abdomen, discoloration, and the reduced lifespan of metamorphosing adult hosts [24,30,36,37] (reviewed in [38]). Experimental infection has demonstrated that DWV alone causes crippled wings [31] and reduced adult lifespan [17], including an acceleration of the temporal polyethism schedule of workers [39,40]. Less is known about the impact of DWV on honey bee pupae [29], although it is reported to cause elevated brood mortality [41].

DWV is arguably the most intensively studied virus of honey bees, from genetic, molecular, and biological characterization [27,42] through to the high-resolution tertiary structure of its virion capsid [43]. These studies refer to the originally described variant of DWV, which is now termed DWV genotype A (DWV-A) [17], and for which a molecular clone has been developed [44]. In 2004, a second major variant of DWV (genotype B) was described from *V. destructor* and associated honey bees [33]. Originally termed *Varroa destructor virus-1*, DWV-B has ca. 84% sequence similarity to DWV-A [33]. Following convention in virology to term variants with >7.5% sequence dissimilarity as distinct genotypes [45], these two DWV variants have been termed: genotypes A and B [46,47,48]. Recombinants between genotypes A and B have often been reported [49,50,51,52,53,54], suggesting that recombination is frequent, or that recombinants have a selective advantage over parental genotypes. DWV has also been considered to represent a quasispecies [27]. Although originally described in the Netherlands [33], DWV-B seems to be increasing in prevalence and distribution, though alternatively it may have gone under-recorded or unrecorded. It has now been reported in Great Britain [17,51,55,56], Belgium [40], the Caucasus [57], France [58], Germany [18], Israel [50], Kenya [59], South Africa [60] and the United States (USA) [61], where sampling in 2010 and again in 2016 suggests that DWV-B has increased rapidly in prevalence. A third variant, which has been termed DWV-C, has been identified [47], and other genotypes of DWV may exist [62]. 

Central to understanding disease emergence is the ability to define the genetic variants of a pathogen in terms of their ability to cause harm to their hosts, which is termed virulence, and is often correlated with their ability to replicate in hosts [63]. We have formerly shown though experimental inoculation that DWV-B has elevated virulence in adult honey bees compared to DWV-A [17], shortening adult lifespan [39,40] and leading to overwinter colony decline during the season when, in temperate regions, colonies are devoid of brood (larvae and pupae) [18]. We have also found that DWV-B replicates faster than DWV-A in adult hosts [17], which is the most parsimonious explanation for why the former is more virulent that the latter in adult hosts. On the contrary, earlier studies have suggested that DWV-B may exclude DWV-A [46], whilst Gisder et al. [64] have recently suggested that DWV-A and DWV-B are more virulent, or replicate most efficiently or to higher titers in host honey bee pupae and vector *V. destructor* mites, respectively. An understanding of the epidemiology of DWV and its genotypes requires knowledge of their comparative impact on honey bee pupae. Firstly, this is because pupae are present in a colony for much of the year (early spring to autumn) [65], and they are likely pivotal in accounting for viral prevalence and the intensity of infection in a colony [66]. Secondly, this is because the major DWV vector *V. destructor* only reproduces on host pupae [38], when it readily transmits virus between hosts [28,29,30].

Using a conceptually simple experimental design coupled with high-resolution next-generation sequence data of viral inocula and resultant host bees, we compared the virulence of DWV-A and DWV-B in metamorphosing honey bees in terms of pupal mortality, wing deformities, and viral titer. We conclude that DWV genotypes A and B differ little in their impact on host pupae, which may help to explain why both have emerged to become major pathogens of honey bees.

## 2. Materials and Methods

### 2.1. Source of Honey Bees

Honey bee pupae for experiments were taken from two colonies (colony one, colony two) in the General Zoology apiary at Martin Luther University Halle-Wittenberg, Germany, which were originally purchased as the subspecies *A. mellifera carnica* and are typical for beekeeping in the region. Both were inspected visually for *V. destructor* mites and by quantitative real-time PCR (qPCR) for eight common viral targets: DWV-A, DWV-B, *Acute bee paralysis virus* (ABPV), *Black queen cell virus* (BQCV), *Chronic bee paralysis virus* (CBPV), *Israeli acute paralysis virus* (IAPV), *Sacbrood virus* (SBV), and *Slow bee paralysis* virus (SBPV), using the primers given in McMahon et al. [56] (for all viruses except CBPV) and [67] (for CBPV) using methods described below (Section 2.4). *Varroa destructor* mites were not seen, and viruses were not detected in colonies at a threshold cycle (Ct) of 35 cycles or less (Ct >35), which is a threshold that minimizes the rate of false positives [68]. 

White-eyed worker pupae were uncapped and carefully removed from frames of brood using fine forceps, and then kept in Petri dishes in incubators at 35 °C and 50% relative humidity (RH) for one to two hours. Pupae that showed signs of damage (blackening) were rejected, as we assumed that we had physically damaged them during removal from their natal cells. Surviving pupae were immediately used for the propagation of genotype-specific DWV inocula or in inoculation experiments.

### 2.2. Virus Propagation and Assessment of Inocula

To propagate a DWV-A inoculum and a DWV-B inoculum for experiments, we injected one μL of the genotype-specific inocula of McMahon et al. (2016), which was originally extracted from an adult, heavily virus-infected honey bee with normal wings from Great Britain (DWV-A) and Germany (DWV-B), into white-eyed pupae taken from another, virus-free colony at the same source apiary as colonies one and two (Figure 1a). After three to five days, pupae were harvested in groups of two to five, crushed in 0.5 M of cold potassium phosphate buffer (PPB pH 8.0), and the resulting homogenate was screened for viruses by qPCR using the methods described below (Section 2.4). We screened our new inocula for the presence of DWV-A and DWV-B as well as six other common honey bee RNA viruses: ABPV, BQCV, CBPV, IAPV, SBV, and SBPV, using the primers described above (Section 2.1). We always generated the correct genotype-specific DWV inoculum from the original inocula [17], although occasional contamination by BQCV caused us to reject a small number of batches of pupal homogenates. As BQCV is at very high prevalence in honey bees [56], it is unsurprising that pupae were occasionally infected with this virus, and that their homogenates were hence rejected by us when generating DWV-inocula. Uninjected white-eyed pupae were treated and screened for viruses in the same way as the pupae that were used to generate viral inocula. As these uninjected pupae were devoid of virus, we used them to generate a control inoculum that was identical to viral inocula, but for the lack of virus. 

For the absolute quantification of DWV-A or DWV-B in our inocula, external DNA standards were generated using primers in [56] (*stand’ curve* primers), which were quantified on an Epoch spectrophotometer, and then used to generate a dilution series (10^−1^–10^−8^) for each DWV genotype that covered the complete range of Ct values in our inocula and our experimental material. Although RNA standards are considered preferable [48], our approach to viral quantification follows the standard methods in bee-virus research [68], and seems robust, because we have found that the results from them reflect those from the next-generation sequence data that was generated using the same viral samples [17]. The dilution series was amplified by qPCR using specific primers for DWV-A or DWV-B to give duplicate standard curves for DWV-A and DWV-B, with our inocula amplified on the same PCR plate. Primer efficiencies were 90–92% for a DWV-A *RdRp* primer pair, and 98–100% for a DWV-B *RdRp* primer pair, and correlation coefficients (R^2^) ≥ 0.988. Our inocula containing only DWV-A or only DWV-B at known concentrations were aliquoted and stored at –80 °C for use in experiments, as was our control inoculum devoid of virus. 

To precisely check the genetic makeup of our DWV inocula, we extracted RNA from them using methods in Appendix B, with the inclusion of a DNAse step. RNA was submitted to commercial mRNA library preparation and ultra-deep next-generation sequencing (NGS) on an Illumina platform (GATC Biotech) as two separate libraries. Sequenced reads were mapped to the *A. mellifera* reference genome (v. 4.5), and to two DWV-A and DWV-B reference genomes (GenBank Accession Numbers NC_004830.2 and NC_006494.1, respectively) using Bowtie 2 [69] (each read was only scored once). Mapped reads were then assigned to *A. mellifera*, DWV-A or DWV-B. Reads that were mapped to both DWV genotypes were counted as DWV-A or DWV-B in the same proportion as the reads that could be uniquely assigned to either DWV-A or DWV-B; they were spread equally across the DWV genome. We then assembled DWV reads using IVA [70] under the default settings of software that is specifically designed for the *de novo* assembly of RNA virus genomes, to generate contigs of DWV. To do so, we used all of the DWV NGS reads mapped by Bowtie 2 (above), and any additional DWV reads identified by the *de novo* assembly of the remaining reads when screening for other viruses, as described in Appendix C. Each contig was then included in a phylogenetic tree using PhyML [71] to visualize its relationship to DWV-A and DWV-B, and identify potential A–B recombinants. Details of the screening of NGS libraries for other viruses, including the detection of full-length *Apis rhadbovirus-1* (BRV-1) [72,73], and other contaminants are given in Appendix C. NGS data files are accessible under BioProject ID PRJNA515220.

### 2.3. Viral Inoculation and Virulence

To test the impact of viral genotype on pupal development, we injected one µL of viral inoculum per white-eyed pupa (Figure 1a) directly into its hemolymph between its second and third abdominal tergites using a Hamilton syringe (hypodermic needle outer diameter: 0.235 mm), simulating viral transmission by *V. destructor*. Viral inocula were firstly diluted in 0.5 M of cold PPB (pH 8.0) to final concentrations of 10^2^ and 10^4^ genome equivalents per µL, as quantified by qPCR. Then, one µL containing 10^2^ or 10^4^ of DWV-A, 10^2^ or 10^4^ of DWV-B, or 10^2^ or 10^4^ of an equal mix of DWV-A plus DWV-B was injected into a pupa. Other pupae were injected with the control inoculum devoid of virus, representing a control group. To avoid cross-contamination, syringes were cleaned after each use, and different syringes were used for inocula A, B, the mix, and control. Additional pupae were kept uninjected to determine the impact of injection *per se* on early pupal survival and wing deformation (Figure 1b). Each treatment group comprised 16 pupae from each of the two source colonies, and every treatment was administered on the same day and by the same person.

Pupae from all of the treatments were placed in 96-well microtiter plates, which were placed vertically so that pupae were horizontal, and maintained in incubators at 35 °C and 50% RH to monitor pupal development. At 24 hours post-inoculation (p.i.), 10.4% (25 of 240) injected pupae had died, whereas zero uninjected pupae (0 of 32) had died. As control-injected and viral-injected pupae died at the same rate in the first 24 hours (difference in proportions, Fisher exact test *p* = 0.793), we assumed that the cause of mortality was physical injury by the hypodermic needle, and not virus inoculum *per se*. Therefore, the pupae that died within 24 hours p.i. were eliminated from further analysis. 

Mortality from day two p.i. through to eclosion was recorded daily as a measure of viral virulence. At day six to day seven p.i., adult workers eclosed. Then, the wing morphology of eclosing adults was inspected visually to determine the impact of virus on wing deformation (Figure 1b), which was another measure of viral virulence. To assess whether viral titer was related to wing deformation or the viral genotype, eclosing adult bees were stored at −80 °C for viral quantification exactly as described above (Section 2.2). 

To check that viral inocula were viable and the virus was replicating following injection, 24 additional pupae were injected as above with 10^2^ or 10^4^ DWV-A or 10^2^ or 10^4^ DWV-B and frozen either on day zero (immediately after injection) or on day three p.i.; then, viral titer was quantified as described above (Section 2.2). Further, in order to test that the virus in the inocula that we had injected into pupae had actually replicated, we again resorted to ultra-deep NGS analysis of one eclosing adult each of a pupa injected with DWV-A (individual pupal code: D4-DWV-A) and a pupa injected with DWB-B (individual pupal code: V4-DWV-B) with the same methods and bioinformatics algorithms used to analyze viral inocula, as described in Section 2.2 and in Appendix C. NGS data files are accessible under BioProject ID PRJNA515220.

### 2.4. RNA Extraction and Detection of Virus

We used standard methods for RNA extraction and cDNA synthesis [68] (and see Appendix B): (i)when testing whether honey bee source colonies were free of virus, (ii)when screening pupal homogenates (our inocula) for RNA viruses and quantifying DWV titer in them,(iii)when quantifying viral titers in adult worker bees arising from inoculation experiments, and(iv)when quantifying viral titers in pupae at zero and three days p.i. 

For viral detection, qPCRs were performed for each sample in a Bio-Rad C1000 thermal cycler (Bio-Rad, Munich, Germany), using SYBRgreen Sensimix (Bioline, Luckenwalde, Germany) and the primers described above (*2.1 Source of Honey Bees*) with the following program: five minutes at 95 °C, followed by 40 cycles of 10 seconds at 95 °C, 30 seconds at 57 °C and 30 seconds at 72 °C. We used a Ct threshold of 35 to define a sample as positive for a virus (Ct < 35). We ran quality control checks on each qPCR 96-well reaction plate to ensure that we had amplified and quantified the correct target (Appendix B).

For absolute viral quantification, duplicate qPCRs were performed for each sample, and the mean Ct value was used. All of the qPCR plates contained a dilution series covering eight orders of magnitude (10^−1^–10^−8^) of an external DNA standard (see Section 2.2), which we used to generate the calibration curves for the quantification of the virus target. qPCRs were repeated for samples whose duplicate Ct values differed by >0.5. 

### 2.5. Statistical Analyses

Generalized linear mixed-effects models (GLMMs) with binomial error structure were used to analyze the overall effect of viral inoculation on bee mortality and wing deformities; Tukey *post hoc* tests were then used for pairwise comparisons between viral treatments. We used GLMMs with Poisson error structure to test whether the wing deformities in emerging adults were associated with the higher viral titer in each treatment. As the focus of our study was on differences between viral genotypes, colony was used as a random factor in all of the mixed-effect models. Additionally, we used generalized linear models (GLMs) with Poisson error structure to test for differences in viral titer between the two DWV genotypes. All of the model analyses were performed using the R package *lme4* v.1.0-6 [74]. *Post-hoc* tests were performed using the R package *multcomp* [75]. If overdispersion was detected in GLMs and GLMMs, we used a quasibinomial and a quasi-Poisson model. All of the model (GLMM and GLM) assumptions were checked visually, and were found to conform to expectations (e.g. normality of the distribution of residuals, homogeneity of variances, linearity). All of the analyses were performed in R v 3.5.0 [76], in which GLM is included in the R base statistics functions. 

## 3. Results

### 3.1. Analysis of Inocula

By qPCR, we detected only DWV-A in the DWV-A inoculum, and only DWV-B in the DWV-B inoculum. When we mapped our NGS data to DWV reference sequences using Bowtie 2, we also found each inoculum to contain its own genotype with very little contamination of the other genotype (see Appendix C and Appendix A). We detected a small amount of *Apis rhadbovirus-1* (BRV-1) [72,73] in the assembled contigs of each inoculum’s NGS dataset (Appendix A), but only a few fragments of the other putative viruses or uncharacterized Eukaryote genes (<0.33%), and none of DWV genotype C, which is another described genotype of DWV [46,47,48]. To test for the impact of BRV-1 in the inocula on our experiment, we analyzed 36 eclosing adults that had been experimentally inoculated as pupae by our inocula using qPCR with the BRV-1 primers and the protocol of [72]. In only one of 36 bees could we detect a positive signal, which was albeit weak (Ct = 32.15), suggesting a low titer of BRV-1 in this individual, and little contamination of our infection experiment by BRV-1. 

To confirm that experimental inocula had replicated in pupae and explore further the possible replication of BRV-1 in our experiment, we subjected two additional experimentally inoculated and eclosing adult bees to ultra-deep NGS analysis exactly as described above as two separate libraries (one library of a DWV-A experimentally inoculated bee, and one library of a DWV-B experimentally inoculated bee). We repeated our NGS data analyses as described above, and revealed each bee to contain the DWV genotype with which it had been inoculated with very little contamination of the other genotype, or of recombinants between the genotypes, or of other putative viruses or uncharacterised Eukaryote genes (see Appendix C and Appendix A). The bee inoculated with DWV-A had a trace of BRV-1 reads (0.000007% of reads), whilst the bee inoculated with DWV-B was completely devoid of BRV-1 reads (Appendix A). Therefore, we considered BRV-1 not to have impacted our experiment; our analyses suggest that BRV-1 may not be pathogenic for *A. mellifera* pupae.

Our IVA-bioinformatics software *de novo* assemblies of the DWV reads from our NGS data revealed only a single full-length genome contig per library, corresponding to DWV-A for inoculum-A (and the pupa inoculated with A), and corresponding to DWV-B for inoculum-B (and the pupa inoculated with B). NGS read coverage of the DWV-A and DWV-B inocula was consistently high across the genome (Appendix A). The DWV-A contig of our DWV-A inoculum differed by 263 bp (263 mismatches, 97.7% sequence identity) and the DWV-B contig of our DWV-B inoculum by 78 bp (77 mismatches and one insertion, 99.3% sequence identity) from the respective GenBank reference sequences (Appendix A). Each genotype varied in sequence by <0.5% across its genome (Appendix A).

Inocula full-length genome contig sequences (Appendix A) differed minimally from those of [17], from which they were derived (one single-nucleotide protein (SNP) difference in DWV-A, three SNP differences in DWV-B; Appendix A). Furthermore, full-length genome contig sequences of viral NGS reads derived from adult hosts differed minimally from those of inocula with which they had been experimentally inoculated as white-eyed pupae (two SNP differences in DWV-A, five SNP differences in DWV-B; see Appendix A). 

In addition to the full-length genome DWV-A or DWV-B contigs that were generated by the IVA assemblies of DWV-A (the DWV-A inoculum and the DWV-A inoculated pupa) and DWV-B (the DWV-B inoculum and the DWV-B inoculated pupa), partial genome contigs were also generated from two of the four NGS datasets. The pupa inoculated with DWV-A (D4-DWV-A) contained four short DWV-B contigs, whilst two short DWV-A contigs were generated from the DWV-B inoculum (Appendix A). These short contigs (Appendix A) mirror the results from mapping the same reads using Bowtie 2 to reference sequences, with a very small number of DWV-B reads in the DWV-A experimental material, and a very small number of DWV-A reads in the DWV-B experimental material (Appendix A). Visual comparison by alignment (Appendix A) and phylogenetic analysis of contig sequences generated by IVA (Appendix A) suggest that our inocula and experimentally inoculated pupae harboured little or no A-B recombinants. These patterns also suggest that the virus we injected into a pupa had replicated and that viral genotype varies little during the course of an infection.

The sequence similarity of the genome of our DWV-A inoculum to the genome of our DWV-B inoculum was 84.4% (1691 mismatches, 20 insertions and deletions), which was a difference that was considerably higher than the genetic variability within an isolate of a genotype (<0.5%, see Appendix A), or of a single genotype across the course of an infection (Appendix A). The two viral inocula did not form a single, interconnected mutant cloud or quasispecies, and remained consistently distinct across the course of a host infection.

### 3.2. Impact of Inoculation on Honey Bee Pupae

In our initial check that viral inocula were viable and the virus was replicating following injection, we did not detect any virus in the pupae by qPCR at day zero p.i. (Appendix A), which was likely because the virus amount was too small to detect. However, at day three p.i., pupae contained high viral titers of the genotype with which they had been injected (DWV-A mean ± SE = 1.16 × 10^12^ ± 1.66 × 10^11^; DWV-B mean = 1.07 × 10^11^ ± 6.21 × 10^10^; Appendix A), indicating that inocula were viable and the virus was replicating well. Additionally, at day three p.i., there were no significant differences in viral titer between pupae injected with either 10^2^ or 10^4^ genome equivalents (quasi-Poisson GLM; DWV-A, z = 0.120, *p* = 0.904; DWV-B, z = -0.742, *p* = 0.458; Appendix A), which was possibly because viral replication had reached an asymptote, and pupae had reached their viral carrying capacity. Henceforth, we combined the data of the two viral quantities (10^2^ and 10^4^ genome equivalents) for each viral treatment. We note that, at day three p.i., DWV-A showed a slightly higher titer than DWV-B (GLM, z = –2.688, *p* = 0.007; Appendix A).

The inoculation of white-eyed pupae with virus led to a significantly higher mortality (mean 29%) compared with control-injected pupae (mean 11%; GLMM; χ^2^ = 4.140, *p* = 0.041; Figure 2). Differences among viral treatments (inocula A, B, and A+B) were minor and statistically insignificant (Tukey *post hoc* tests; B *versus* A: z = -0.307, *p* = 0.759; AB *versus* A: z = 0.041, *p* = 0.968; AB *versus* B: z = -0.346, *p* = 0.730; Figure 2), suggesting that both A and B genotypes of DWV have an equal and subtle effect in elevating pupal mortality by 18% above control. We note that the random term ‘colony’ was significant in these analyses (likelihood ratio test; χ^2^ = 9.739, *p* = 0.001), which was likely because virus-induced pupal mortality was lower in colony two compared to colony one (Appendix A). 

For the bees that survived through to eclosion, 83% of the virus-injected bees possessed one or more deformed wings, whereas only 23% of the control-injected bees exhibited deformed wings (Figure 1b), which was a statistically significant difference (GLMM; χ^2^ = 36.202, *p* < 0.001; Figure 3). Differences in the proportion of eclosing bees with wing deformities among viral treatments (inocula A, B, A + B) were small and statistically non-significant (Tukey *post hoc* tests; B *versus* A: z = –1.345, *p* = 0.179; AB *versus* A: z = 0.631, *p* = 0.528; AB *versus* B: z = 1.883, *p* = 0.060; Figure 3). The random term ‘colony’ was non-significant in this analysis (likelihood ratio test; χ^2^ = 1.238, *p* = 0.266); bees from both colonies suffered equally from wing deformations (Appendix A).

Bees eclosing with normal wings did not differ in viral loads from those eclosing with deformed wings for each of the three virus treatment groups (A, B, A+B; quasi-Poisson GLMM, DWV-A treatment with normal *versus* deformed wings, z = 0.152, *p* = 0.879; DWV-B treatment with normal *versus* deformed wings, z = 0.180 *p* = 0.238; DWV A+B treatment [total viral load] with normal *versus* deformed wings, z = −0.186, *p* = 0.852; Figure 4). Control-injected bees had only background titers of virus (Figure 4). For the DWV A+B treatment, the viral loads for each of the genotypes also did not differ between bees eclosing with normal wings *versus* those eclosing with deformed wings (quasi-Poisson GLMM treatment DWV A+B; DWV-A z = −0.139, *p* = 0.889; DWV-B z = −0.199, *p* = 0.842; Figure 4). This pattern did not change when analyzing the viral load solely by viral genotype rather than by experimental treatment (Appendix A). Therefore, wing deformation was independent of viral load or viral genotype under our experimental paradigm. 

DWV-B replicated to a higher titer in eclosing bees that had the DWV-B treatment than DWV-A did in eclosing bees that had the DWV-A treatment (quasi-Poisson GLMM; DWV-A *versus* DWV-B: z = 3.713, *p* < 0.001; Figure 4). The difference in titer was modest, ca. twofold (DWV-A mean 1.20 × 10^12^; DWV-B mean: 2.68 × 10^12^).

We note that the DWV-B titer in the adults that had received DWV A+B treatment and successfully eclosing with normal wings was slightly although significantly lower than the DWV-A titer in the same bees (quasi-Poisson GLMM; DWV-A *versus* DWV-B: z = −5.155, *p* < 0.001; Figure 4). However, differences in titer were small (Figure 4), and the result is based on only *n* = 2 successfully emerging bees. Therefore, we caution against the over-interpretation of this result. 

## 4. Discussion

Using a controlled laboratory experiment, we show that DWV genotype B is no different in virulence for host honey bee pupae than DWV genotype A; both genotypes caused subtly elevated pupal mortality, and both caused considerably elevated wing deformities in eclosing adult workers. However, in single-genotype infections, DWV-B did replicate to a slightly higher titer than DWV-A in pupae that reached adulthood. 

McMahon et al. [17], using genetically near-identical inocula to those employed in our experiments, found that DWV-B was more virulent than DWV-A when injected into honey bee adults, replicating faster and killing adults sooner than DWV-A. Although one observational study has suggested that DWV-B may be less virulent than DWV-A [46], the higher virulence of DWV-B has since been independently supported by others through the experimental infection of adult honey bees in Belgium [40]. In contrast to our findings, a recent experimental study [64] found that DWV reduced in virulence following a single passage through a host pupa, and that the viral variant switched from DWV-B to DWV-A in passing through a second pupa. Our experimental data did not show the same result, and we did not observe an oscillation between genotypes after passaging the virus through host pupae. Experimental paradigms differed between the study of Gisder et al. [64] and our own, making it difficult to reconcile differences in results, although more in-depth bioinformatics analysis of viral inocula and infected host honey bees might help interpret them. 

DWV A–B recombinants have often been reported in honey bees infested by *V. destructor* mites [49,50,51,52,53,54]. In additional, an A–B recombinant has been found to be more virulent than DWV-A alone, replicating to higher titers when vectored by *V. destructor* mites into pupae [52]. Although Ryabov et al.’s [52] study did not compare the virulence of A or A–B recombinants with B, these studies [17,40,52] nevertheless provide experimental support for the view that DWV-B, or A–B recombinants, are more virulent than DWV-A in adults. Our simultaneous inoculation of pupae with both DWV-A plus DWV-B did not lead to greater viral impact on hosts, compared to DWV-A alone or DWV-B alone. This suggests that viral recombinants may not differ in virulence from parental genotypes in our study, or that the recombinant virus was generated very late in the course of our experiment; thus, putative recombinants with high virulence remained at low titer and had insufficient time to exert an effect on hosts. McMahon et al. [17] similarly found that an inoculum of both DWV-A plus DWV-B was no different in virulence from DWV-B alone in adults, although both showed elevated virulence over DWV-A. These data leave the role of viral recombination between genotypes for viral virulence in DWV open.

We did not find elevated virulence of DWV-B over DWV-A in pupae. Given the higher virulence of B over A in adults that we have formerly detected using related inocula to those employed here [17], the lack of difference in the virulence of DWV-A *versus* DWV-B in pupae is surprising. We found that DWV-A replicated to slightly higher titers than DWV-B after three days, but that DWV-B replicated to even higher titers than did DWV-A by the time of adult emergence when each was experimentally injected alone into white-eyed pupae. However, DWV-B neither increased pupal mortality nor did it induce greater wing deformities in eclosing adults compared with DWV-A. We hypothesize that the short duration of pupal development (six to seven days from white-eyed pupa to the eclosion of an adult worker honey bee) may have been insufficient for differences in pupal virulence among viral genotypes to become apparent in our experiment. For example, McMahon et al. [17] found that adult mortality in response to DWV experimental infection was most pronounced only at 10 to 20 days p.i.. *Varroa destructor* mites naturally enter host brood cells during the fifth larval instar stage; immediately after sealing of the brood cell, they commence feeding on the host, which is when they potentially transmit virus during the host’s prepupal stage (review in [38]). This would provide a ca. 11-day time window for DWV replication in honey bee pupal hosts [34] (p. 9). To test our hypothesis that our experiment did not extend over a long enough period of time to allow differences in virulence between genotypes A and B of DWV to become apparent, one could inoculate prepupae immediately after cell sealing and follow the pupal fate for longer than in our experiment, which terminated after six to seven days because pupae had already eclosed. However, this experiment would be technically challenging, because prepupae are extremely delicate and difficult to manipulate (pers. obs.).

Alternatively, the virulence of DWV-B in pupae may indeed be no different from that of DWV-A. As the main route of transmission of DWV among honey bees is through the vector *V. destructor* mite [27], which reproduces on host pupae within sealed brood cells, pupal-induced mortality would lead to the failure of mite reproduction and limit viral transmission. This effect would select for the reduced pupal virulence of all of the DWV genotypes [66], and might explain why we found no difference in pupal virulence (mortality, wing deformities) between DWV-A and DWV-B. In support of this view, mathematical models suggest that Acute bee paralysis virus (ABPV), which is another virus transmitted by *V. destructor* between pupae, may be disfavored by *V. destructor* transmission because its virulence is too high in pupae [77,78]; it effectively eliminates itself from host colonies by killing the host and vector before viral transmission is possible. Indeed, our data suggest that both DWV-A and DWV-B possess traits (low virulence in pupae) that allow them potentially to increase in prevalence in honey bee populations, underpinning their emergence across the globe wherever *V. destructor* has invaded *A. mellifera* populations [58]. 

We found that the pupae from one colony suffered lower mortality than those of the other colony. These data suggest that genetic variance may exist among honey bees in their tolerance to DWV, which is possibly mediated through differences in innate immunity and antiviral defense [79]. Although our experiments were based on pupae from only two colonies, and were not designed to test for genetic variance in the host response to DWV, our results suggest that the topic warrants further study.

DWV injected into white-eyed honey bee pupae has been previously shown to induce wing deformities in resultant adults in a dose-dependent manner [31]. The log-linear relationship they [31] described between viral dose and the deformed wing phenotype predicts between 28% (for a dose of 10^2^ DWV genome equivalents) and 59% (for a dose of 10^4^ DWV genome equivalents) wing deformities for our experimental protocol, which is remarkably close to the 60% elevation in wing deformities we actually recorded. Notwithstanding this dose–response relationship, an open question remains over why DWV does not always induce deformed wings. DWV can be detected around wing buds during pupal development [34] (pp. 109–114), suggesting that its presence at this site may play a role. As primordial wing buds are laid down early in the larval development of holometabolous insects [80], we hypothesize that wing deformation may depend upon the presence or concentration of DWV in pupal wing buds or in unfolding wings during a critical stage or stages of metamorphosis. However, our data do suggest that viral genotype (A *versus* B) is not related to wing deformation in eclosing adults, which is a view shared by Brettell et al. [81].

It has been suggested that the invasion of *V. destructor* into honey bee populations not only led to the emergence of DWV [19,20], but that it also may have selected for a particular DWV variant favored by mite vectoring [31,66,82]. DWV-B is a likely candidate variant favored by *V. destructor* transmission. Although first described in 2004 from the Netherlands [33], it is nowadays widespread and at high prevalence in Great Britain [17,55,56] and Germany [18]. It may have formerly been unrecorded or under-recorded, and therefore, its current high prevalence in Europe may be an artefact through a lack of detection in earlier studies. However, DWV-B has increased dramatically in prevalence since its first detection in the USA in 2010 to becoming almost as prevalent as DWV-A across the country in 2016 [61]. Phylogenetic analysis also suggests the rapid population expansion of DWV-B [58]. Our data on pupal virulence support the idea that DWV-B has biological traits allowing it to emerge in honey bees parasitized by *V. destructor*, where it has a major impact on adult honey bee health [17,40] and overwinter survival [18] because of its greater virulence in adult hosts compared to DWV-A. The higher titers of DWV-B in comparison to DWV-A, which is markedly so in adults but also subtly so in pupae, as we show here, also support the idea that DWV-B is emerging, leading to a gradual mixing of genotypes and potentially the generation of recombinants, or even the replacement of DWV-A. If DWV-B is better able to be transmitted by, or even replicate in, *V. destructor*, as suggested by Gisder et al. [64], this might also facilitate its emergence. The potential for the replacement of DWV-A by DWV-B will require additional information on competitive interactions in co-infected hosts, including recombination [52] and the fate of the recombinant virus.

## Figures and Tables

**Figure 1 viruses-11-00114-f001:**
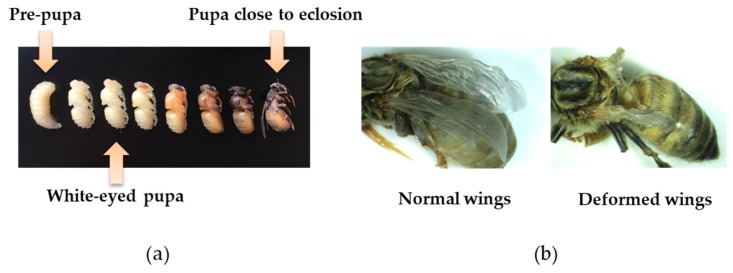
(**a**) White-eyed pupa used to generate deformed wing virus (DWV) inocula and in inoculation experiments, with prepupae to the left, and more advanced pupal stages to the right (each stage separated by ca. 24 hours); (**b**) left, normal winged adult; right, adult with deformed wings.

**Figure 2 viruses-11-00114-f002:**
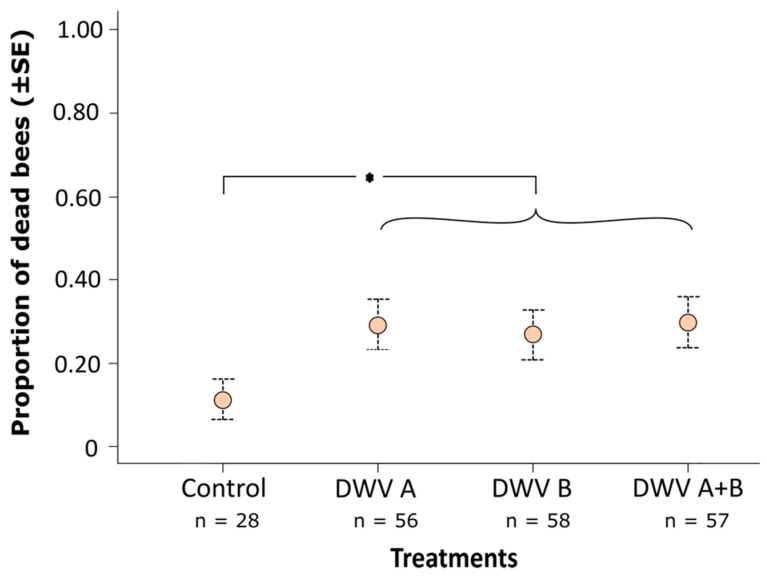
Mean proportion of dead honey bee pupae inoculated at the white-eyed stage with a control solution (virus-free extract of bees), deformed wing virus A (DWV-A) inoculum, deformed wing virus B (DWV-B) inoculum, or simultaneously DWV-A and DWV-B inocula, and dying before or at eclosion. DWV-inoculated bees suffered significantly higher mortality (mean 29%) than control bees (mean 11%; GLMM; χ^2^ = 4.140, *p* = 0.041) but mortality did not differ between virus-inoculated treatments (*, Tukey *post hoc* pairwise comparisons, all *p* > 0.05); *, *p* < 0.05.

**Figure 3 viruses-11-00114-f003:**
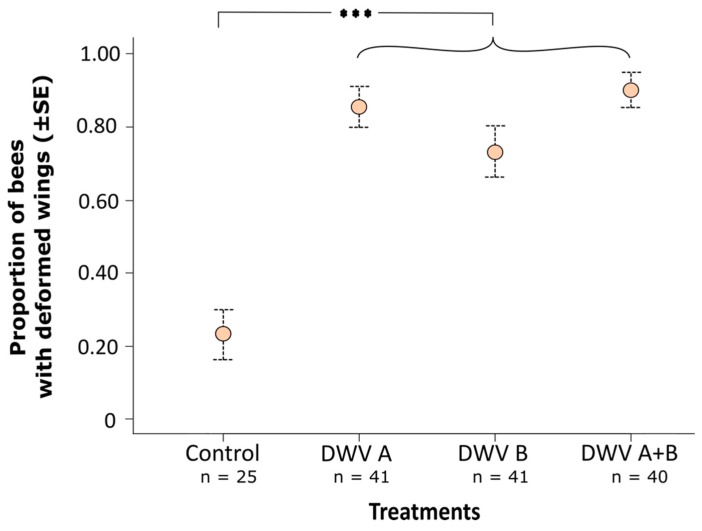
Mean proportion of successfully eclosing honey bee adults with deformed wings that as white-eyed pupae had been inoculated with a control solution (virus-free extract of bees), DWV-A inoculum, DWV-B inoculum, or simultaneously DWV-A and DWV-B inocula; virus-inoculated bees more frequently eclosed with wing deformities (mean 83%) than control bees (mean 23%; GLMM; χ^2^ = 36.202, *p* < 0.001), but the probability of wing deformation did not differ between virus-inoculated treatments (Tukey *post hoc* pairwise comparisons, all *p* > 0.05); ***, *p* < 0.001.

**Figure 4 viruses-11-00114-f004:**
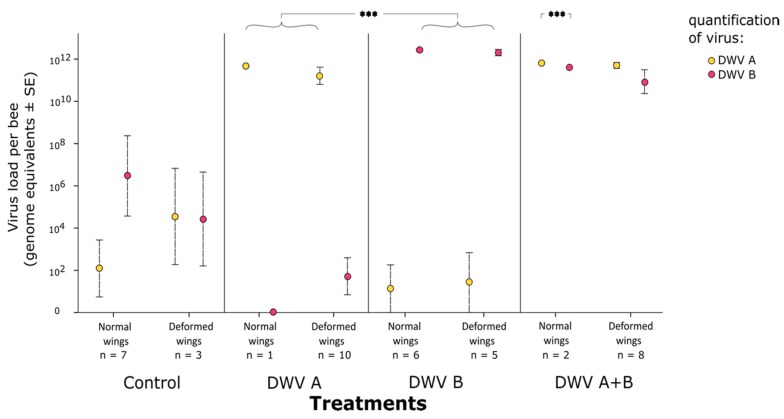
Mean DWV titers in eclosing honey bee adults that as white-eyed pupae had been inoculated with control (virus-free pupal extract), DWV-A inoculum, DWV-B inoculum, or simultaneously DWV-A and DWV-B inocula. Viral titers did not differ within a viral treatment for bees emerging with normal *versus* deformed wings (GLMM *p* > 0.05). Bees inoculated with DWV-B had a higher titer of DWV-B than the bees inoculated with DWV-A had of DWV-A (GLMM *p* < 0.001), which was a pattern that was reversed for the bees simultaneously inoculated with DWV A+B that emerged with normal wings (GLMM *p* < 0.001); ***, *p* < 0.001.

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
