# Peer review of "The Two Prevalent Genotypes of an Emerging Infectious Disease, Deformed Wing Virus, Cause Equally Low Pupal Mortality and Equally High Wing Deformities in Host Honey Bees"

_viruses, 2019, doi:10.3390/v11020114_

Round 1
Reviewer 1 Report
The paper provides results of carefully designed and well-executed study aimed to compare
infection of two distinct variants of Deformed wing virus, DWV-A and DWV-B (which is also
known as Varroa destructor virus-1, VDV1) in the honey bee pupae. Artificial injection of the
honeybee pupae models natural transmission of these DWV strains by its Varroa mite vector,
which is known to lead to severe honey bee health problems, which manifests itself with
development of overt symptoms, such as “deformed wings”, and may result in colony death.
Previously published papers suggested that DWV-B has higher virulence than DWV-A and
associated with increased colony losses. Surprisingly, despite the importance for honey bee
health and agriculture, there is a paucity of studies which systematically investigated mortality
and symptom development in the honey bees infected with DWV-A and DWV-B. The presented
study addressed some of these questions by quantitatively comparing impact of these strains of
DWV on the honey bees in individual and in a mixed infections.
In particular, similar mortality levels and proportions of the bees with deformed wings were
observed in the groups infected with either DWV-A, DWV-B, or the mixture of both DWV variants.
Importantly, high levels of DWV-A and DWV-B were found in some injected bees with normal,
non-deformed, wings. This study also showed that DWV-B replicated to significantly higher levels
than DWV-A in the case of single infections, although, surprisingly, in the mixed infection DWV-A
accumulated to higher levels than DWV-B. These results provided novel insight into DWV
pathogenesis, symptom development and interaction between strains of this economically
important virus.
*****************
Reply: We thank the referee for their positive and encouraging comments.
*****************
Minor comments
Line 23.
Remove "relatively recently", first sequences of both strains were published within two
years, DWV-A (Kakugo virus strain - 2004, PA strain - 2006) and DWV-B - 2004.
*****************
Reply: We removed ‘relatively recently’ and following words to make sense. It is true that full
genome sequences of the two genotypes appeared at approximately the same time, yet partial
sequences of DWV-A have abounded in the literature for many years before 2004 whilst only in
2004 were sequences of DWV-B reported, including a complete genome sequence. We therefore
feel it is appropriate to retain the words, ‘originally described’ for genotype A.
*****************
Line 26.
remove "healthy"
*****************
Reply: We removed ‘healthy’.
*****************
Line 157
PPB - is it " potassium phosphate buffer " ?
*****************
Reply: In our first use of ‘potassium phosphate buffer’ (revised ms, line 146), we erroneously put
‘PBB’ in brackets after its use, which we have now corrected. PPB is therefore explained in full on
its first use and, thereafter (including original ms, line 157), we use PPB.
*****************
Lines 196- 206.
Specify which enzyme was used for Reverse Transcription; was cDNA primed using random or
specific primers?
*****************
Reply: These details were already given in Appendix A. We now move our citation of Appendix A
to the start of the Material and Methods section 2.4. The relevant section of Appendix A read (and
still reads) (revised ms, lines 629-30): cDNA was synthesised from RNA extracts using oligo(dT)18
primers (Thermo Scientific) and M-MLV and Revertase (Promega, Mannheim, Germany)
following the manufacturer's instructions.
*****************
Figure 2 and Figure 3
Marks showing proportions of mortality and deformed wing bees should have the same colour.
*****************
Reply: The marks now have the same colour.
*****************
Figure 4, Lines 313-318.
Indicate in the figure legend that yellow marks show quantification of DWV-A and red marks -
quantification of DWV-B.
*****************
Reply: A small legend has now been added to the top right of the figure, showing A-yellow and Bred.
*****************
Line 319
Give the fold difference between DWV-A and DWV-B accumulation levels.
*****************
Reply: The difference is 2.2-fold (DWV-A mean 1.20 x 1012; DWV-B mean: 2.68 x 1012) We now
state this (revised ms, line 450).
*****************
Line 321-322
"replicates faster" -> "replicates to higher levels" - to assess dynamics several time points had to
be assessed,
*****************
Reply: This is a good point. We have deleted the sentence.
*****************
Lines 339-351.
Include references on higher incidence of DWV-VDV1 recombinants in Varroa-infested honey
bees in the UK (Moore et al.,2012 J. Gen. Virol. 92: 156-161) and France (Dalmon et al.,2017
Scientific Reports 7, 41045). Reference Zioni et al (No. 53) could be cited in this section as well.
*****************
Reply: We have added (revised ms, lines 480-1): ‘and A-B recombinants have often been
reported in honey bees infested by V. destructor mites (and cited Zioni et al. 2011, Moore et al.
2012, and Dalmon et al. 2017).’ We also cited two additional, relevant articles: Wang et al. 2013
and Cornman 2017, as well as re-citing Ryabov et al. 2014. We have also cited these papers in
the Introduction, where we raise the issue of recombination, in response to the issue of
recombination by referee 2.
*****************
_____________________________________________________________________________
Reviewer 2 Report
The article is similar to the lead author’s previous work (McMahon et al. 2016), which compared
the pathogenicity of the two dominant variants of DWV, type A and B. In this instance, the authors
focus on the virulence on honeybee pupae, instead of adults. The premise of the paper is a good
one, and the methodologies and results are, on the whole, scientifically sound and well
presented, although more could of been done with the bioinformatics and sequencing data (more
on this later).
*****************
Reply: We have added to the bioinformatics analysis of our NGS datasets (our detailed response
to the specific point is below).
*****************
The paper concludes that similar incidences of deformed wings were attributed to both variants (A
or B), and the overt symptom was not directly related to viral load. The paper found that both
variants resulted in similar mortality of inoculated pupae. I believe these results to be scientifically
sound, although they are a little overshadowed by a very recent publication (Gisder et al. 2018)
which was able to tease apart this complicated relationship by showing that the source of the
virus can significantly alter the resulting virulence. For instance, virulence of an inoculum from a
deformed bee was attenuated after one passage through pupae. These results suggest that the
observed differences in virulence which have been noted by different groups is more complex
than solely which variant is present, and recognised that DWV exists as a quasispecies made up
of several master variants. Nevertheless, the results from this paper are worthy of publication, as
every little piece of the puzzle is useful for the field. However, it would be more helpful to the
reader if the results were discussed in the context of these more recent findings.
*****************
Reply: The Gisder et al. (2018) paper came out just before we submitted our ms so we apologise
that we did not see it before our submission. We have now read the paper and cite it in our
revised ms, pointing out in our introduction (revised ms, lines 95-7) that Gisder et al. (2018)
suggest switching of viral variant between A and B as DWV is transmitted between host bee and
mite vector. In our revised discussion (revised ms, lines 473-9), we again refer to Gisder et al.’s
(2018) main premise, that of switching of viral variant between A and B, and state clearly that our
results are in disagreement with theirs. Given that experimental paradigms differ between these
two experiments (Gisder et al.’s and ours), it would be rash of us to make any more of the
differences in results; Gisder et al. (2018) took virus from multiple bees with deformed wings then
passaged it sequentially through two pupae whereas we passaged virus from bee to bee to bee.
We have now suggested in our discussion (revised ms, lines 473-9) that an in-depth
bioinformatics analysis of Gisder et al.’s (2018) material might help resolve differences because,
currently, Gisder et al. (2018) hypothesise multiple viral variants within a host yet they only
provide a consensus sequence from each inoculum/infected bee NGS dataset (their data would
allow them to test their own hypothesis, but they do not do so) and they do not provide the raw
NGS data to allow others to test their hypothesis e.g. the issue of viral recombination is missing
from Gisder et al. (2018).
The reviewer is correct in saying that DWV, as all RNA viruses, exhibits intra-host genetic
diversity, and the survival measures reflect the phenotype of a cloud of related mutants present in
the inoculum and the host. In response to this point, we already added in the supplementary
material of the original submission two figures (Supplementary Material Figures S2A and B)
illustrating the level of variation captured with NGS data along the viral genomes for DWV-A and
DWV-B in our experiment, which we estimated as<0.5% (given in the Results (revised ms, line
354) and in the legend to Figure S2). Therefore, we are confident that the results presented in the
current manuscript represent the phenotypes of the two genotypes with little or no contamination
of one by another genetic variant. Moreover, the viral genome assemblies requested by the
reviewer (see below) support the view that both genotypes were kept separate in our
experimental work and were devoid of recombinants.
*****************
The introduction could do a better job of introducing the DWV quasispecies, and the complicated
interplay between variants e.g. recombination. Whether DWV exhibits quasispecies dynamics is
seen as controversial in this field (for reasons beyond me). If the authors do not support this
argument, they should at least address it, and present evidence for and against. However, there
are now several papers which provide very good evidence for the quasispecies dynamic (Gisder
et al. 2018; Mordecai, Brettell, et al. 2016; Mordecai, Wilfert, et al. 2016; Dalmon et al. 2017).
*****************
Reply: We address quasispecies first and recombination second.
Quasispecies: Quasispecies dynamics is an evolutionary theory that, in short, implies that natural
selection acts on the cloud of mutants rather than on a single viral haplotype, which may
paradoxically be fitter than any and all mutants of the cloud. This is similar to group level
selection, also coined the ‘survival of the flattest’, in opposition to the Darwinian concept of
‘survival of the fittest’. Whether RNA viruses, such as DWV, evolve under quasispecies dynamics
remains controversial. As mentioned by an excellent opinion paper (Holmes 2010), the difficulty is
unambiguously to show that natural selection favour several haplotypes with variable fitness
rather than one fittest viral haplotype. This is obviously beyond the topic of our work, because in
the current manuscript we describe the phenotype of two distinct strains of DWV. However, we
agree with the reviewer that our introduction could raise the quasispecies concept, which we now
do in our revision (revised ms, lines 79-80). Also in the discussion, where we now address the
results of Gisder et al. (2018) that are used to support the quasispecies concept for DWV, we
additionally state that our data contradict those of Gisder et al. (2018) (revised ms, lines 473-7).
However, we refrain from suggesting our data argue against the quasispecies concept for DWV
because our experiment was not designed to – and cannot - test the concept of quasispecies
applied to DWV.
Recombination: Given the empirical evidence for recombination and its potential for generating
novel, epidemiologically relevant viral variants, this is an important issue, especially for
understanding DWV epidemiology. We now raise recombination among DWV genotypes in the
introduction and again, more substantively, in the discussion (also in response to reviewer 1’s
related comment; see revised ms, lines 77-9 and 480-1, respectively).
*****************
For the sequence data analysis, a little work is required. Rather than a read mapping approach of
the assembly of the DWV genomes, the NGS analysis should be repeated using a de-novo
analysis pipeline designed for RNA viral quasispecies (e.g. Vicuna, IVA). I am aware that this was
done for the unmapped reads, but it would be useful on the DWV reads too. The read mapping
method used could easily miss a novel variant or recombinant. Just because a read maps to a
reference, this does not imply it is identical. The resulting contigs should then be phylogenetically
analysed to show which variant they represent, rather than just putting the sequence in the
supplementary file. This is not useful for the average reader.
*****************
Reply: This is a very good point, which we have now addressed fully by reanalysing our NGS
datasets. We thank the reviewer for their insight into viral NGS analysis and the suggestion to
improve our analysis. We have now generated de novo assemblies of all four NGS datasets
(inoculum A, inoculum B, an adult inoculated as a pupa with A, an adult inoculated as a pupa with
B) using IVA. We describe the relevant bioinformatics steps in the revision (end of Materials and
Methods section 2.2) and present an alignment of the assembled contigs in a new Supplementary
Material (Figure S5). We have then phylogenetically analysed the reads and presented a
phylogeny of them (new Figure S6). These IVA assemblies (all contigs now given in Figures S3
and S4) show:
inoculum A: a single contig of full-length DWV-A without recombinant contigs
pupa injected with inoculum A: a contig of full-length DWV-A and 4 short contigs of partial DWVB,
without recombinant contigs
inoculum B: a contig of full-length DWV-B and 2 short contigs of partial DWV-A, without
recombinant contigs
pupa injected with inoculum B: a single contig of full-length DWV-B without recombinant contigs
The phylogeny (new Figure S6) shows that all DWV-A contigs are very similar and cluster tightly
together with reference DWV-A sequences in one branch whilst all DWV-B contigs are very
similar and cluster tightly together with reference DWV-B sequences in another branch of the
phylogeny.
These data reflect very closely the read counts we gave in Supplementary Material Tables S1
and S2, in which inoculum A and the pupa injected with inoculum A contained primarily A and
inoculum B and the pupa injected with inoculum B contained primarily B. We additionally update
Table S3, giving the sequence similarity across our new and reference sequences, using the new
IVA-generated full length genome contig sequences.
We now make all our NGS data files publicly available in the SRA (short read archive) of NCBI,
accessible under the BioProject ID PRJNA515220. These details are added to the ms.
*****************
There is little information on how the authors looked for recombination within the sequencing
data, this would be of especial interest in the competitively challenged samples. There is little
information on how the consensus sequences were made, or how much variation existed around
this consensus.
*****************
Reply: We originally had not specifically searched for recombinants in our NGS datasets but
rather relied on sequence coverage across the genome to suggest the presence of recombinants.
We have now used IVA to generate de novo assemblies of all four NGS datasets and examined
resultant contigs for recombinants. Visual inspection (alignment) of the IVA contigs revealed only
DWV-A or DWV-B but no recombinant contig. A phylogeny of those IVA contigs (new Figure S6)
also shows that contigs do not contain a recombinant.
In our initial submission, we mapped all NGS reads to DWV reference sequences using Bowtie 2
and then generated consensus sequences from mapped DWV reads, which is the approach we
followed in McMahon et al. (2016). We now replace these consensus sequences with IVA
contigs, as recommended (see our response to the immediately preceding point). The new, fullgenome
IVA contigs differ from our 1st submission consensus sequences by 0 and 5 bases for
DWV-A (inoculum and pupa/adult) and by 4 and 4 bases for DWV-B (inoculum and pupa/adult)
i.e. our former (consensus) and our new (de novo contig) approach generate near-identical
sequences.
Figure S2 gave the genetic variability around the full genome sequences, which is<0.5%.
We agree that it would be interesting to examine recombination in competitively challenged
pupae. To do so, we have recently set up new experimental infections in which we challenge
pupae with both A and B, but we are a long way from generating results from them.
*****************
More information is needed about the inocula, what was the disease state of the bees from which
it was isolated. Put this into the context of (Gisder et al. 2018)) recent findings.
*****************
Reply: We now add this information as (revised ms, lines 143-144):
originally extracted from an adult heavily virus-infected honey bee with normal wings from Great
Britain (DWV-A) and Germany (DWV-B)
The adult honey bees from which we extracted virus were heavily infected (high titre) with DWV,
though their wings were normal. Gisder et al. (2018) collected virus from multiple adult deformedwing
honey bees. We can concur that the disease status of the bees from which we extracted
virus was ‘low’, and the colonies were ‘unhealthy’ (Natsopoulou et al. 2017). The disease status
of the host honey bees from which Gisder et al. (2018) extracted virus was also ‘low’.
Unfortunately, Gisder et al. (2018) did not report on the proportion of emerging pupae with
deformed wings in their experiment to know whether the status ‘deformed wings’ varies across
their viral extracts. In our experiment, it did not. Moreover, we found that, when experimentally
injecting a bee with DWV (regardless of DWV genotype), viral titre was high after 3 days,
continued to increase till eclosion of the adult, and was independent of whether the resultant adult
has deformed wings or not. ‘High titre’ is therefore likely a much better measure than ‘deformed
wings’ of whether a honey bee adult is healthy or not, in the experimental paradigms we and
Gisder et al. (2018) used. We think this diminishes the argument that adult honey bees with
deformed wings contain a different virus (variant) than adult honey bees with normal wings –
when both have very high viral titres. We can go on to hypothesise that, in a colony with, say, 60
adults with deformed wings, there will be another ca. 40 honey bees with normal wings that have
equally high viral titres and of low health status as the deformed-wing honey bees (the proportion
of virus-inoculated honey bees with deformed wings beyond the control treatment in our
experiments was 60%).
*****************
The observed difference between colonies suggest that host immunity could be a factor. Or other
factors? Perhaps this could be mentioned more in the discussion,
*****************
Reply: We have now added a small paragraph to the discussion (revised ms, lines 534-8) to
suggest that our results (on genetic variance in tolerance to DWV) warrant further investigation.
We include the idea that host immunity might play a role, though refrain from going into deeper
discussion of this result because we did not set out to test for genetic variance (our experiment
was not designed to do so). We hope we can thereby encourage others to consider this issue
further. The text now reads (revised ms, lines 534-8):
We found that pupae from one colony suffered lower mortality than those of the other colony. These
data suggest that genetic variance may exist among honey bees in their tolerance to DWV, possibly
mediated through differences in innate immunity and antiviral defence (Brutscher et al. 2015). Though our
experiments were based on pupae from only two colonies and were not designed to test for genetic variance
in host response to DWV, our results suggest the topic warrants further study.
*****************
The discussion is let down a little by some sweeping statements which do not appear to be
supported by the results and tend to oversimplify some complicated issues. For example, the
gradual emergence of type B to replace type A. As this paper shows that the virulence is similar,
it is a bit of a stretch. Instead, it would be more useful if the authors examine the literature, over
time, which in some cases suggest that type B is increasing in prevalence in the USA (Ryabov et
al. 2017), whilst other research found that type A is selected for by Varroa (Martin et al. 2012).
Try to lay out all sides of the argument so that the reader can make an informed decision. I think
this would be more persuasive to argue your case.
*****************
Reply: We understand the issue and, rather than trying to force one hypothesis (take-over of
DWV-A by DWV-B), we now rephrase our language by indicating what we have shown versus
what others have shown, by hypothesising there might be a take-over of B by A, and by
suggesting alternative scenarios (that B might have gone un- or under-recorded). We hope that
are revised language comes across as more even-handed.
The Martin et al. (2012) study does not address the take-over of A by B, but rather shows that,
when varroa mites are present in a colony, the DWV titre shoots up, which the scientific
community largely understands and accepts as a product of very efficient transmission by varroa
mites of DWV. Interestingly, the genetic variant of DWV that shoots up in titre within a colony
differs from colony to colony (Supplementary material of Martin et al. 2012). DWV-B was
apparently not detected in Hawaii in Martin et al.’s study so it is not possible to use it to address
take-over of A by B (supplementary materials of Martin et al. 2012 and pers. comm., Martin and
Brettell, 2017).
*****************
Smaller changes:
Title: The title is a little clunky in terms of its sentence structure. The facilitating viral spread part
of the title is a little bit misleading, as this is not really shown in the paper. I suggest you remove
this part and make the similar observed virulence the focus of the title.
*****************
Reply: Though our paper was submitted to the special issue of Viruses on ‘viral emergence’, for
which we wrote the title, we agree that we do not really show emergence, other than arguing in
the last paragraph of the discuss that DWV-B might be considered an emerging viral pathogen.
As recommended by the referee, we have removed the final phrase of the title and deleted a few
superfluous words.
*****************
Line 49: Mention the change of host of V. destructor.
*****************
Reply: This has now been mentioned at this point in the ms, and included with a complimentary
phrase, now moved from the following sentence (revised ms, line 51).
*****************
Line 49: Long sentence, needs work.
*****************
Reply: One phrase has been moved to the former sentence (see point above) and the sentence
split in two to improve its clarity (revised ms, lines 54-7).
*****************
Intro should include mentiuon of this important paper (Lamp et al. 2016)
*****************
Reply: We now include reference to the DWV molecular clone generated by Lamp et al. 2016 and
cite the paper (revised ms, line 73).
*****************
Line 65-76. Should talk about recombination between variants.
*****************
Reply: We now add text at this point to highlight that A-B recombinants have often been reported,
with associated citations. This also follows the recommendation of referee 1 (revised ms, lines
77-9).
*****************
Line 77. Should at least mention the presence of a third variant (type C) (Mordecai, Wilfert, et al.
2016). And likely others (see (Roberts, Anderson, and Durr 2017), supplementary figures 2 and
3).
*****************
Reply: Thank you for pointing out the supplementary figures of Roberts et al. 2017. We now
mention the third variant DWV-C at this point in the ms, the potential for other variants, and we
cite these papers. We do so at the end of the former paragraph, which is more appropriate
because it deals with the variants of DWV (revised ms, lines 85-6).
*****************
Line 79-80. The lead authors cites their own paper, and states that it is unequivocal evidence.
The issue of the pathogenicity of different variants is not as straightforward as claimed, and there
is varying evidence to suggest this. For example, the recombination between variants (Moore et
al. 2011), resulting in a variant better adapted to transmission by Varroa. Type A was selected by
Varroa in Hawaii (Martin et al. 2012), and this has been shown to be true in many other studies.
How do we address the superinfection exclusion of B over A in apparently healthy
colonies (Mordecai, Brettell, et al. 2016). What about the host immune response to each virus? It
is tempting to oversimplify, but not helpful in the long run.
*****************
Reply: We understand that the reviewer wishes us to present a more even-handed view of
virulence. We have therefore modified the entire paragraph, clearly indicated that it is our
(experimental) results which support a point and now present the alternative results of others’
(largely observational) studies (revised ms, lines 87-117).
*****************
Line 118. Confusing sentence structure. Why not just say “We screened our new inocula for the
presence or absence of DWV-A and DWV-B” etc… Screening for the absence of something
implied bias.
*****************
Reply: We have now deleted the unnecessary, additional details and merely stated that we
screened for the given viruses (revised ms, lines 147-149).
*****************
Line 126. Change “was” to “were”.
*****************
Reply: We changed ‘was’ to ‘were’ and simplified further parts of the sentence to improve clarity
(revised ms, lines 155-7).
*****************
Line 127. Remove semi-colon and start new sentence to improve clarity.
*****************
Reply: We removed the semicolon and rephrased the sentence to improve clarity (see point
immediately above, too).
*****************
Line 132: The authors should at least recognise the limitations of using DNA standards compared
to RNA standards (Kevill et al. 2017).
*****************
Reply: We have added a caveat that RNA standards might be considered better, plus a
justification for our use of DNA standards (that they are robust because our data generated using
them is well reflected in our next generation sequence data derived from the same samples)
(revised ms, lines 174-8).
*****************
Line 135/203 Avoid using PCR as a verb as in qPCRed.
*****************
Reply: We apologise for the use of this short-hand. We now modified the first use of PCR as a
verb by changing to ‘amplified by qPCR’ and we deleted its second use, which was superfluous.
We checked the rest of the ms and did not find additional uses of PCR as a verb (revised ms, line
178/line 259).
*****************
Line 160: change was to were
*****************
Reply: We changed the verb to ‘were’ and modified the sentence so that the plural form was
correct (revised ms, lines 217-218).
*****************
Line 162: change was to were
*****************
Reply: As above, we changed the verb to ‘were’ and modified the sentence so that the plural form
was correct (revised ms, lines 217-9).
*****************
Line 165: delete “to pupae”
*****************
Reply: We deleted ‘to pupae’ and changed ‘each’ to ‘every’ to improve clarity (revised ms, lines
221-2).
*****************
Line 223. Inconsistent use of passive and active voice.
*****************
Reply: We have reworded the sentence and maintain the active voice for clarity (revised ms,
lines 290-291).
*****************
Line 352: implied bias is not useful.
*****************
Reply: We here presented one of our main results in the first sentence of this paragraph and, in
the rest of the paragraph, we tried to explain why we did not find the result in the current
experiments with pupae that one might predict from our former experiments with adults (that
DWV-B is more virulent than DWV-A). We did not intend any bias and, re-reading, we do not see
any bias, unless one wishes to question our former results (which we think are robust). We have
nevertheless revised the text to highlight that our current results are surprising in the light of our
former results. By doing so, we hope we have removed any bias that the referee perceived
(revised ms, lines 494-6).
*****************
Brutscher, L.M.; Daughenbaugh, K.F.; Flenniken, M.L. Antiviral defense mechanisms in honey bees.
Current Opinion in Insect Science 2015, 10, 71-82, doi: 10.1016/j.cois.2015.04.016
Cornman, R.S. Relative abundance of Deformed wing virus, Varroa destructor virus 1, and their
recombinants in honey bees (Apis mellifera) assessed by kmer analysis of public RNA-seq data.
J. Invertebr. Pathol. 2017, 149, 44-50, doi: 10.1016/j.jip.2017.07.005
Dalmon, A., C. Desbiez, M. Coulon, M. Thomasson, Y. Le Conte, C. Alaux, J. Vallon, and B. Moury.
2017. “Evidence for Positive Selection and Recombination Hotspots in Deformed Wing Virus
(DWV).” Scientific Reports 7 (January): 41045.
de Miranda, J.R.; Genersch, E. Deformed wing virus. J. Invertebr. Pathol. 2010, 103, S48-S61, doi:
10.1016/j.jip.2009.06.012
Gisder, Sebastian, Nadine Möckel, Dorothea Eisenhardt, and Elke Genersch. 2018. “In Vivo Evolution
of Viral Virulence: Switching of Deformed Wing Virus between Hosts Results in Virulence
Changes and Sequence Shifts.” Environmental Microbiology, November.
https://doi.org/10.1111/1462-2920.14481.
Holmes, E.C. The Evolution and Emergence of RNA Viruses. Oxford University Press: Oxford, UK,
2009; p 288.
Holmes, E.C. The RNA virus quasispecies: fact or fiction? J. Mol. Biol. 2010, 400, 271-273, doi:
10.1016/j.jmb.2010.05.032
Kevill, Jessica, Andrea Highfield, Gideon Mordecai, Stephen Martin, and Declan Schroeder. 2017.
“ABC Assay: Method Development and Application to Quantify the Role of Three DWV Master
Variants in Overwinter Colony Losses of European Honey Bees.” Viruses 9 (11): 314.
Lamp, Benjamin, Angelika Url, Kerstin Seitz, Jürgen Eichhorn, Christiane Riedel, Leonie Janina Sinn,
Stanislav Indik, Hemma Köglberger, and Till Rümenapf. 2016. “Construction and Rescue of a
Molecular Clone of Deformed Wing Virus (DWV).” PloS One 11 (11): e0164639.
Martin, S. J., A. C. Highfield, L. Brettell, E. M. Villalobos, G. E. Budge, M. Powell, S. Nikaido, and D. C.
Schroeder. 2012. “Global Honey Bee Viral Landscape Altered by a Parasitic Mite.” Science 336
(6086): 1304–6.
McMahon, Dino P., Myrsini E. Natsopoulou, Vincent Doublet, Matthias Fürst, Silvio Weging, Mark J. F.
Brown, Andreas Gogol-Döring, and Robert J. Paxton. 2016. “Elevated Virulence of an Emerging
Viral Genotype as a Driver of Honeybee Loss.” Proc. R. Soc. B 283 (1833): 20160811.
Moore, Jonathan, Aleksey Jironkin, David Chandler, Nigel Burroughs, David J. Evans, and Eugene V.
Ryabov. 2011. “Recombinants between Deformed Wing Virus and Varroa Destructor Virus-1 May
Prevail in Varroa Destructor-Infested Honeybee Colonies.” The Journal of General Virology 92 (Pt
1): 156–61.
Mordecai, Gideon J., Laura E. Brettell, Stephen J. Martin, David Dixon, Ian M. Jones, and Declan C.
Schroeder. 2016. “Superinfection Exclusion and the Long-Term Survival of Honey Bees in
Varroa-Infested Colonies.” The ISME Journal 10 (5): 1182–91.
Mordecai, Gideon J., Lena Wilfert, Stephen J. Martin, Ian M. Jones, and Declan C. Schroeder. 2016.
“Diversity in a Honey Bee Pathogen: First Report of a Third Master Variant of the Deformed Wing
Virus Quasispecies.” The ISME Journal 10 (5): 1264–73.
Natsopoulou, M.E.; McMahon, D.P.; Doublet, V.; Frey, E.; Rosenkranz, P.; Paxton, R.J. The virulent,
emerging genotype B of Deformed wing virus is closely linked to overwinter honeybee worker
loss. Scientific Reports 2017, 7, 5242, doi: 10.1038/s41598-017-05596-3
Roberts, John M. K., Denis L. Anderson, and Peter A. Durr. 2017. “Absence of Deformed Wing Virus
and Varroa Destructor in Australia Provides Unique Perspectives on Honeybee Viral Landscapes
and Colony Losses.” Scientific Reports 7 (1): 6925.
Ryabov, Eugene V., Anna K. Childers, Yanping Chen, Shayne Madella, Ashrafun Nessa, Dennis
vanEngelsdorp, and Jay D. Evans. 2017. “Recent Spread of Varroa Destructor Virus-1, a Honey
Bee Pathogen, in the United States.” Scientific Reports 7 (1): 17447.
Wang, H.; Xie, J.; Shreeve, T.G.; Ma, J.; Pallett, D.W.; King, L.A.; Possee, R.D. Sequence
recombination and conservation of Varroa destructor virus-1 and Deformed wing virus in field
collected honey bees (Apis mellifera). PLoS ONE 2013, 8, e74508, doi:
10.1371/journal.pone.0074508
Wilfert, L.; Long, G.; Leggett, H.C.; Schmid-Hempel, P.; Butlin, R.; Martin, S.J.M.; Boots, M. Deformed
wing virus is a recent global epidemic in honeybees driven by Varroa mites. Science 2016, 351,
594-597, doi: 10.1126/science.aac9976
Zioni, N.; Soroker, V.; Chejanovsky, N. Replication of Varroa destructor virus 1 (VDV-1) and a Varroa
destructor virus 1–Deformed wing virus recombinant (VDV-1–DWV) in the head of the honey bee.
Virology 2011, 417, 106-112, doi: 10.1016/j.virol.2011.05.009